# Standardization of Molecular MRD Levels in AML Using an Integral Vector Bearing ABL and the Mutation of Interest

**DOI:** 10.3390/cancers15225360

**Published:** 2023-11-10

**Authors:** Boaz Nachmias, Svetlana Krichevsky, Moshe E. Gatt, Noa Gross Even-Zohar, Adir Shaulov, Arnon Haran, Shlomzion Aumann, Vladimir Vainstein

**Affiliations:** Department of Hematology, Hadassah Medical Center and Faculty of Medicine, Hebrew University of Jerusalem, Jerusalem 91200, Israel; krichevsky@hadassah.org.il (S.K.); rmoshg@hadassah.org.il (M.E.G.); noagros@gmail.com (N.G.E.-Z.); ashaulov@gmail.com (A.S.); arnonharan@gmail.com (A.H.); saumann@gmail.com (S.A.); vladimir.dr@gmail.com (V.V.)

**Keywords:** acute myeloid leukemia, measurable residual disease, plasmid base

## Abstract

**Simple Summary:**

Acute Myeloid Leukemia is the most common leukemia in adults and has a dismal prognosis. An allogeneic bone marrow transplant provides the best curative approach. However, due to its related morbidity and mortality, the decision is based on risk assessment at diagnosis and response to therapy. Accurate assessment of molecular Minimal Residual Disease (MRD) has provided a powerful tool to assess the depth of response and risk of relapse. However, validated and standardized molecular MRD is currently limited to typical NPM1 mutations and core-binding factor translocations. To allow for a better standardization of other identified mutations, we constructed a vector that bears both the sequence for the mutation of interest and the ABL1 gene, thus allowing us to calculate the mutation copy number using the inherent ABL1 gene standards. We have implanted this approach in several identified mutations including atypical NPM1, IDH1/2 and RUNX1.

**Abstract:**

Quantitative PCR for specific mutation is being increasingly used in Acute Myeloid Leukemia (AML) to assess Measurable Residual Disease (MRD), allowing for more tailored clinical decisions. To date, standardized molecular MRD is limited to typical NPM1 mutations and core binding factor translocations, with clear prognostic and clinical implications. The monitoring of other identified mutations lacks standardization, limiting its use and incorporation in clinical trials. To overcome this problem, we designed a plasmid bearing both the sequence of the mutation of interest and the ABL reference gene. This allows the use of commercial standards for ABL to determine the MRD response in copy number. We provide technical aspects of this approach as well as our experience with 19 patients with atypical NPM1, RUNX1 and IDH1/2 mutations. In all cases, we demonstrate a correlation between response and copy number. We further demonstrate how copy number monitoring can modulate the clinical management. Taken together, we provide proof of concept of a novel yet simple tool, which allows in-house MRD monitoring for identified mutations, with ABL-based commercial standards. This approach would facilitate large multi-center studies assessing the clinical relevance of selected MRD monitoring.

## 1. Introduction

Treatment of Acute Myeloid Leukemia (AML) has evolved in recent years with the incorporation of novel agents and Measurable Residual Disease (MRD) as a tool to assess remission depth. Patient and disease characteristics currently dictate the treatment approach including intensive versus non-intensive regimens, referral to an allogenic bone marrow transplant (allo-BMT) and the combinations of novel agents [1]. Response assessment by regular flow cytometry with a cut-off of 5% blasts is limited by its ability to detect a low level of disease, which would translate to a high rate of relapse. MRD evaluation either by flow cytometry [2] or by mutation-based analysis [3] has proved a powerful tool in the post remission induction period to evaluate deep anti-leukemia therapy efficacy, detection of early relapse, consolidation assignment (e.g., selection of transplant vs. no-transplant strategy as well as conditioning intensity), and as surrogate end-point in clinical trials [4,5,6,7,8]. Data clearly show that patients who achieve MRD negativity have a significantly longer progression free survival (PFS) and overall survival (OS) [9]. Current efforts are aimed to further study the ability of MRD to direct treatment decisions, with promising results [10,11,12], and to better identify the correct “MRD eraser”, such as in the INTERCEPT trial [13].

Techniques for molecular monitoring of MRD such as quantitative PCR for both gDNA and RNA targets, digital droplet PCR for known hotspots in gDNA and next generation sequencing (NGS) MRD monitoring are evolving; however, they have still not been widely accepted [7,14,15]. Quantitative PCR (qPCR) KITs for routine diagnostics and MRD follow-up of fusion oncogenes expression and DNA point mutation/deletion/insertions are widely available and contain a serial of plasmid standards for quantification. Unfortunately, all those available KITs are designed for detection of specific but common changes. Currently, standardized qPCR tests are only available for typical *Nucleoplsmin 1* (*NPM1*) mutations (type A, B and D) and core-binding factor translocations (CBF). Outstanding mutations different from the most frequent changes are not covered by commercial KITs. For example, about 12% of NPM1 positive patients have non-A, or B, or D mutations and have to be followed by homemade methods [16].

For most leukemia-associated genes, such as *IDH1* and *IDH2*, *FLT3*, *WT1*, *CEBPα*, *RUNX1*, *DNMT3A* and *TET2*, qPCR is widely available but lacks standardization. Notably, some of these mutations are not useful for molecular follow-up due to uninformative character [17,18], unstable expression or technical difficulties [3,19]. In other genes, such as *RUNX1* and *CEBPα* with VAF < 50%, individual mutations are very good and stable markers of assessing the persistence of the leukemic clone. Those mutations can be detected by sequence screening procedures, like the Sanger sequencing of a specific gene, different types of NGS, or by using special NGS panels [20]. Either DNA or RNA-based qPCR can be used, with RNA-based analysis being usually more sensitive as multiple RNA copies (versus only two DNA copies) are present per cell.

In centers where un-standardized qPCR MRD evaluation is performed, serial dilutions of the sample from diagnosis is used to build the calibration curve and to determine the relative MRD level in log reduction units [21]. Additionally, in RNA-based techniques reference gene expression (usually *ABL1*) is required for normalization as well as a control of cDNA quantity and quality [22,23]. This method falls short as it does not allow an accurate absolute MRD level determination (in copy number units) and it can be used for a limited number of times during the follow-up as the lab gradually runs out of the RNA/DNA extracted from the diagnostic initial sample.

To overcome these obstacles and allow for a wide and available standardization, we developed a vector bearing the synthetic minigene including sequences of the mutation of interest and the *ABL1* gene. Using commercially available *ABL1* standards, we can quantify in copy number our individual synthetic minigene by included the *ABL1* sequence that allows, as well, matched copy number determination of the mutation of interest. Thus, it offers a sensitive and quantitative tool for MRD assessment. We provide first-to-date proof of concept for this approach in MRD assessment for atypical *NPM1* mutation, RUNX family transcription factor 1 (*RUNX1*) and Isocitrate dehydrogenase (*IDH*) *1/2* mutations.

## 2. Materials and Methods

### 2.1. Patients

All newly diagnosed AML patients since 2018 in our institution have been screened for atypical *NPM1*, IDH1/2 and *RUNX1* mutations and positive cases were followed by individualized MRD monitoring. These mutations were selected due to their relative frequency, stability of expression and as they are not considered pre-leukemic mutations, which might persist with questionable prognostic value [3,21,24]. Patients’ response was assessed after every treatment line/cycle, until response or referral to allo-BMT, and at specific follow-up time points for patients in remission. Response assessment based on the detection of a patient’s specific mutation by individually designed PCR assay was correlated with standard of care evaluation, including bone marrow (BM) morphology, flow cytometry, and, for patients after allo-BMT, with short tandem repeats (STRs) for detection of donor chimerism.

### 2.2. Mutation Detection

Total white blood cells (WBC) were extracted from BM and peripheral blood (PB) samples: red blood cells were lysed with red blood cells (RBC) lysis buffer, WBC washed with PBS and cell pellet subjected to DNA and RNA preparation. The Maxwell^®^ RSC Blood DNA KIT (Promega, Madison, WI, USA) and Maxwell^®^ RSC simply RNA Blood Kit (Promega, Madison, WI, USA) were used for DNA and RNA extraction with a Maxwell^®^ RSC Instrument (Promega, Madison, WI, USA). cDNA was prepared from 2-µg RNA using the High-Capacity cDNA Reverse Transcription Kit (cat#AB-4374966 Thermo Fisher Scientific, Waltham, MA, USA). DNA was screened for presence of mutations with Sanger sequencing or Archer^®^ VariantPlex^®^ myeloid panel (Boulder, CO, USA). All found mutations were verified at cDNA level.

### 2.3. Measurable Residual Disease Monitoring

RNA was extracted from PB and BM patients’ samples and reverse transcribed to cDNA. All MRD evaluations were performed on the cDNA. Target gene mutant allele quantification and MRD monitoring were performed with allele specific Real Time PCR (ASqPCR). MRD value was calculated as the number of copies of the target mutated gene per 10^5^ copies of the reference gene, defined as copy number (CN) [25]. An individual PCR system was developed for each patient. The system includes individual TaqMan expression assay with mutation specific primers and probe for ASqPCR, and individual plasmid constructs. Plasmid synthetic individual standards (pSIS) contains a fragment of patient mutation-specific sequence(s) and a fragment of reference gene sequence. Copy numbers of mutation-specific and reference sequences per plasmid molecule are equal. We used commercial *ABL1* standards [22] (Ipsogen cat#674791) to calculate *ABL1* reference copy number of the individual plasmid and patient sample. Five serial dilutions of pSIS minigene were introduced in two separate PCR reactions: one for quantitation of *ABL1* reference copy number and the second for quantitation of mutation in gene of interest. Both reactions were proceeded simultaneously. *ABL1* copy number of each pSIS dilution was calculated with commercial *ABL1* standards. Copy numbers of *ABL1* reference sequences and individual mutation were equal for every pSIS dilution. Therefore, calculated by commercial standards, *ABL1* copy numbers were used to prepare standard curve and calculate copy numbers of patient-specific mutation (Figure 1).

### 2.4. Linearity and Limit of Detection

PCR dynamic range and assay sensitivity were calculated per individual target sequence [26]. Both parameters are very dependent on mutation and surround sequences. For some mutations, we used rh-primers to decrease background amplification. Primers and probes are provided in Appendix A. Individual plasmids were linearized, tested for specificity and sensitivity, and stored at −20 °C/−70 °C as aliquots. New plasmid aliquots were used for standard curves preparation in every PCR run. Five points of serially tenfold diluted individual plasmid in duplicates were run simultaneously in two separate PCR reactions: one for the target and the second for the reference gene. Reference gene copy number was calculated for every plasmid dilution point according to the commercial reference gene standards (Ipsogen cat#674791) and then introduced as copy number of target gene standard curve. Our calculation method was validated by quantification of the plasmid construct with *NPM1* type A mutation and ABL1 reference gene sequences versus commercial standards for *NPM1* mut A cDNA MutaQuant standards (Ipsogen cat# 677591) and *ABL1* Control Gene 4 standards (Ipsogen cat#674791) (Figure 1). Standard curves for both target and reference copy numbers were prepared with the same threshold and only reactions with efficiency in the range of 92–110% and R^2^ > 0.99 were presumed valid. Cut-off positivity was determined as a threshold cycle (CT) value that was one cycle above the value of wild type control samples amplified at a cycling threshold of 0.05. Follow-up cDNA samples were tested in 6 replicates for 45 cycles in an ASqPCR reaction. The reaction was performed with AmpliTaq Gold proofreading DNA polymerase in total volume of 20 µL and carried out on a StepOnePlus instrument (Applied Biosystems). Samples were scored as positive when a minimum of three positive replicates were detected at CT above CT determined as positive cut-off. PCR run include mutation specific standard curve from serial dilution of patient specific plasmid, *ABL1* standard curve from *ABL1* ipsogen standards, 2 wild type cDNA controls (6 replicates for each control sample) and 2 blank templet control. In some cases, relative MRD change post-therapy and during follow-up were calculated by ∆∆Ct method with correction for PCR efficiency [27]. The expression ratio of mutant transcript (as a target gene) versus ABL1 reference gene at diagnosis served as 100% baseline value in single mutation plasmids and as copy number in integral mutation-ABL plasmid.

We follow the ELN consensus document on MRD. MRD test positivity by qPCR is defined as a cycling threshold < 40 in ≥2 of 3 replicates. [3].

The retrospective data collection was reviewed and approved by the local ethics committee (Hadassah Medical Center, Jerusalem, Israel), approval number 009520 in accordance with the Helsinki declaration standards. Written informed consent was exempt given the retrospective collection of the data.

## 3. Results

### 3.1. Generation and Validation of Plasmid-Base PCR for MRD Monitoring

Commercial standards for quantification of AML MRD are available only for *NPM1* type A, B, and D as well as CBF translocations. We developed individual plasmid constructs for MRD detection by ASqPCR, thus generating plasmid synthetic individual standards (pSIS) (Figure 1). The sequence with patient-specific mutation (in genes *NPM1*, *RUNX1*, *IDH1/2*) and the *ABL1* reference sequence were synthesized and cloned in pMA-RQ plasmid backbone to insert the relevant individual mutated gene sequence (*NPM1*, *RUNX1*, *IDH1/2*) with the *ABL1* sequence in a 1:1 ratio (Figure 2a). Copy number of these sequences per plasmid molecule are equal, allowing direct calculation of individual mutations’ copy number according to reference gene copies detected with commercial standard curve for *ABL1*.

To validate our ability to determine the gene of interest copy number, we initially prepared the plasmid containing the type A *NPM1* mutation sequence, allowing us to utilize commercially available standards for both *NPM1* and *ABL1*. We then determined the *ABL1* and *NPM1* copy number in manually prepared, serially diluted aliquots of *NPM1* type A pSIS vector. As shown in Figure 2b,c, there was a very high concordance in copy number measurements of *NPM1* and *ABL1*, thus confirming our ability to use *ABL1* standards to determine the copy number of the gene of interest.

Serial dilutions of either synthetic gene-containing plasmid or patient’s diagnostic sample were used for the calibration curve generation. Next, we evaluated MRD status for the same patient samples by either ∆∆CT method (with efficiency correction or copy number quantification using the diagnostic sample for standard curve generation) or by quantification with pSIS with *ABL1* standards. To compare results obtained by those methods, we converted data obtained with pSIS into percent from diagnosis. Results were mostly concordant (Figure 3). The quantitative range was determined by the difference between the upper margin (disease burden at diagnosis) and lower margin (the PCR lower limit of detection) (Figure 4). For example, the quantitative range of the IDH2R140Q mutation is 10^−3.25^, while for NPM1 mutation it was 10^−5^.

### 3.2. Patient Characteristics and Mutation Profile

Patients’ demographics, disease characteristics and clinical data are presented in Table 1. Mutation specific plasmids were generated for 19 patients: 8 patients with RUNX1 mutations, 5 patients with atypical (non-A, -B, -D type) NPM1 mutations, 5 patients with IDH2 mutations and 1 patient with IDH1 mutation. One patient (#17) had concomitant RUNX1 and IDH2 mutations, with the latter used for MRD monitoring, and two patients had concomitant RUNX1 and IDH1 mutations, of which one of them has RUNX1 plasmid and the other has IDH1 plasmid (#6 and #19, respectively). In patients with more than one mutation, all mutations were initially explored for MRD monitoring potential, and the system with the highest sensitivity was further used.

We observed a general correlation between response assessment by flow cytometry, STR in relevant cases and MRD monitoring in all patients, but MRD positivity could still be detected in cases with 100% donor STR and no excess of blasts. We present detailed data on MRD measurements for selected patients to highlight the significance of MRD monitoring in clinical management (Table 2, Table 3 and Table 4).

#### 3.2.1. RUNX1 Monitoring

Eight patients with *RUNX1* mutations were monitored for MRD. Six received therapy with hypomethylating agent, with or without venetoclax, and two received aggressive chemotherapy with 7 + 3 protocol.

Patient #1 was diagnosed with myelodysplastic syndrome (MDS) with no excess of blasts. He started treatment with azacitidine with reduction in *RUNX1* burden from 30,814 CN at diagnosis, to 548 after one cycle, and demonstrated improvement in peripheral blood counts. Six months later, a decrease in blood counts was noted, and bone marrow demonstrated a rise in *RUNX1* levels to 1064 CN, with no excess of blasts. Pevonidestat was added to the regimen, with *RUNX1* CN reduction to 364 after two cycles, and partial counts recovery. The patients continued treatment with stable blood counts but passed away 6 months later due to refractory multiple myeloma.

Patients #2 and 3, though treated with curative intent for AML, were not found to be eligible for intensive chemotherapy at diagnosis, therefore azacitidine-venetoclax (aza-ven) was given as induction. Patient #2 showed marrow blasts decreased from 40% to 10% after one treatment cycle, without change in *RUNX1* level. As the patient had FLT3 mutation, treatment was changed to Gilteritinib. After one cycle of Gilteritinb, the patient achieved CR by flow, but did not have significant change in the level of *RUNX1*. After the second cycle, the *RUNX1* level still had not changed, prompting referral to an allo-BMT. Gilteritinib was resumed after transplant and in a recent six months follow-up, the patient iss still in CR, with 100% STR and low positive *RUNX1* MRD in BM (Table 2, Figure 5). Patient #3 also achieved MRD positive CR after allo-BMT, with BM aspiration 4 months after transplant showing 1.8% blasts, 100% STR and *RUNX1* reduction from 149,776 at diagnosis to 68 CN. Nine months post-transplant, while with 3.2% marrow blasts by flow, RUNX1 levels increased to 8209, and STR decreased to 33%, prompting donor lymphocyte infusion (DLI) (Figure 5).

Patients #4 was refractory to 7 + 3 induction and therefore received salvage therapy with high dose cytarabine and mitoxantrone (HAM protocol). She was then referred to allo-BMT and achieved MRD negative CR. Patient #7 did not achieve CR after induction therapy with 7 + 3 + midostaurin, and therefore received fludarabine, cytarabine, idarubicin, and granulocyte colony-stimulating factor (G-CSF) with venetoclax (FLAG-Ida-Ven) as salvage protocol and achieved CR by flow with *RUNX1* MRD positive pre-transplant. After allo-BMT, her *RUNX1* was undetectable. Both patients continue to remain in molecular remission on post-transplant follow-up (Table 2, Figure 5).

Patient #5 is a 92-year-old female, diagnosed with AML and started on an aza-ven treatment regimen. She achieved CR by flow after first cycle, with a reduction in *RUNX1* levels from 400,965 at diagnosis to 23,508 CN. In a recent assessment of response after 12 cycles of treatment, the patient is in CR with no detectable *RUNX1* in PB.

#### 3.2.2. NPM1 Monitoring

We report five patients with atypical (non-A, -B, -D) *NPM1* mutations, of which four received aggressive chemotherapy and one aza-ven regimen.

Three patients with atypical *NPM1* mutations achieved morphologic remission after induction therapy with either intensive chemotherapy (#9 and 10), or aza-ven regimen (#11), but still had measurable MRD above the lower detection limit. Patient #9 eventually achieved remission with low positive *NPM1* MRD level (45 CN) after three cycles of high dose cytarabine (HIDAC) consolidations and was referred to follow up with no allo-BMT. Three months later, an increase in MRD to 245 CN was demonstrated (with no marrow blast excess). A repeated BM one month later demonstrated a further increase in *NPM1* to 36,000 CN, and the patient was referred to allo-BMT. Though there was no evidence of morphologic relapse, the patient was treated with two cycles of aza-ven as a bridge to transplant. After two cycles, a 3-log reduction in the MRD level was achieved, the patient entered allo-BMT and remains MRD negative in 3 months post allo-BMT follow-up (Table 2, Figure 5). Patient #10 had morphologic CR with a less than 2-log reduction in MRD after induction and positive MRD (239 *NPM1* CN) after one HIDAC consolidation. She then received one cycle of aza-ven with positive MRD (79 CN), and therefore was referred to allo-BMT at this point. She remains in CR with 100% STR and MRD negative at 12 months post-allo-BMT follow-up (Table 2). Patient #11 was referred to allo-BMT in MRD positive CR after aza-ven induction and achieved MRD negativity post-transplant.

Patients #12 and #13 received 7 + 3 +/− midostaurin induction, achieving CR by flow cytometry post induction. Concordantly, they achieved molecular response of 49 and 216 CN post induction, and negative MRD and 45 CN post consolidation, as compared to 3,221,400 and 1,414,578 CN at diagnosis, respectively. They were both referred to follow up with bone marrow evaluation every 3 months. Patient #13 remains in CR at 24 months follow-up with low positive *NPM1* levels (less than 100 CN, Table 3). Patients #12 was in MRD negative CR for 2 years, then at 24 months follow-up she demonstrated a 1.5-log increase with 314 *NPM1* CN, but no rise in marrow blasts. At 30 months follow-up, *NPM1* CN rose to 98,000, and there was still with no concordant rise in marrow blasts. At 33 months, *NPM1* CN was 463,000, and marrow blasts were 10%. At 37 months, the NPM1 CN was 828,000, with frank relapsed leukemia (26% marrow blasts). The patient received Flag-Ida-Ven salvage regimen followed by allo-BMT in MRD-negative CR. As of last follow up, 9 months post-transplant, the patient is in CR with no marrow blasts, 100% STR and negative MRD (Table 4).

#### 3.2.3. IDH1/2 Monitoring

We present five patients with *IDH2* mutations and one with *IDH1* mutation. All but one patient (#15) received aza-ven therapy.

Patient #15 did not achieve CR after induction with 7 + 3. Following salvage with FLAG-Ida-Ven, the patient reached CR by flow with 2-log reduction in *IDH2* (967 CN). The patient was referred to allo-BMT and showed further improvement achieving negative MRD at 3 months post-transplant (Table 2).

*IDH1/2* MRD monitoring was useful in patients treated with non-intensive regimens as well. Patient #14 demonstrated a morphologic response with 3.7% marrow blasts after one cycle of aza-ven treatment, but *IDH2* levels remained very high (79,000 CN). After the second cycle, *IDH2* levels declined to 3500 CN, with 1.8% marrow blasts, but after the third cycle, *IDH2* levels increased again to 10,500 N, with only minor elevation in marrow blasts (3%). After six cycles of treatment, the *IDH2* level continued to rise to 132,000 CN, although marrow blast count remained low. Morphologic relapse came very soon after, and the patient died of active disease.

Patient #16 achieved MRD negativity after one cycle of aza-ven but had detectable low positive MRD after the second cycle (106 CN). She underwent allo-BMT and showed MRD negativity 3 months post-transplant and remains in ongoing MRD negative remission at one year post-transplant follow-up (Table 2).

Patient #17 achieved morphologic CR after three aza-ven cycles, but *IDH2* levels remained high at 57,000 CN. In the presence of morphologic response and in absence of available therapeutic alternatives, he continued treatment with the same regimen. Currently, the patient remains in CR with continual reduction in *IDH2* levels, as demonstrated by a value of 1450 IDH2 CN in the last test performed after 18 cycles of treatment (Table 3).

Patient #18 presented with AML secondary to essential thrombocytosis, and started aza-ven regimen, with MRD negativity achieved after two cycles of therapy, and ongoing MRD negative remission after allo-BMT (Table 2).

Patient #19 is the only patient in our cohort with *IDH1* mutation MRD follow-up. He is in ongoing morphological remission 6 months into aza-ven therapy with low positive *IDH1* level (1000 CN).

## 4. Discussion

Accurate assessment of remission depth by highly sensitive molecular methods has gained wide clinical use in AML management, allowing both better risk stratification regarding consolidation decisions and longitudinal monitoring to detect and respond early to molecular relapse. Currently, validated, and standardized molecular MRD evaluation is limited to qPCR transcript determination in CBF translocations and type A, B and D *NPM1* mutations [22]. NGS-based MRD assessment methods are evolving but are still not widely available and lack good standardization [28]. Standard amplicon-based or capture-based NGS techniques allow for detection of mutations at 3–5% VAF, making them not suitable for MRD monitoring. UMI-based error-corrected NGS has better sensitivity but still generally up to 0.1% VAF, i.e., ~1 log less than qPCR-based techniques [29]. While having the advantage of assessing a variety of genes in a single test, this type of NGS requires the use of expensive equipment and highly specialized bioinformatics tools, both being not widely available [30].

The identification of molecular- based MRD of *NPM1* or CBF translocation was validated as a poor prognostic factor in several large studies [4,5,31,32]. The translation of MRD information to clinical decision still requires large studies incorporating MRD-adaptive response. Still, *NPM1* MRD positive patients after induction and consolidation are largely regarded as an indication for allo-BMT, whereas MRD negativity, can in certain patients (e.g., favorable, high risk for transplant, *FLT3*-mutated) allow close monitoring without proceeding to transplant. As *NPM1* mutations and CBF translocation encompass up to 30% of AML patients, most patients do not have a valid molecular MRD. Moreover, to conduct large international studies of “in-house” qPCR of other identified mutations, standards are required in order to compare results. To address this problem, we present in the current study a simple approach to utilize the commercially available standards for *ABL1* to assess the copy number of any identified mutation. By generating a plasmid that bears both the sequence of the mutation of interest and the *ABL1* sequence, we were able to correlate the qPCR of the patient sample with a standardized transcript copy number. We provide proof-of concept of this approach with *IDH1/2*, atypical *NPM1* and *RUNX1* mutations in 19 patients with AML. We further demonstrate how this approach of in-house molecular MRD provides a tool to guide clinical decisions of treatment augmentation.

The generation of individual patient specific primers for specific mutation is an attractive approach to better determine response and guide clinical decisions, as implemented in B/T cell receptor rearrangement monitoring in acute lymphoblastic leukemia [23]. This approach has several caveats. First, clonal variations under treatment pressure can significantly change the allelic frequency of specific mutations (e.g., *FLT3*), not necessarily reflecting disease state. Similarly, pre-leukemic clones or germ line mutations associated with leukemia development may still be detected, with no prognostic significance [24]. Again, this emphasizes the need for large studies to validate the prognostic significance of specific mutations [18]. Second, the qPCR sensitivity of various mutations differs, with some providing a highly sensitive marker for MRD assessment (up to 10^−6^), while others have low sensitivity, and thus cannot reliably be used for MRD monitoring. Nonetheless, in-house qPCR-based molecular monitoring for specific mutations is relatively easy to perform and was shown to help guide clinical decisions [21]. To date, in-house of MRD monitoring used ∆∆CT with efficiency correction, and reports were in log reduction compared to diagnosis. As such, achieving a log reduction of 3–4 was partially dependent on the quantitative range (QR), meaning how many log reductions relative to the diagnosis can be detected in an individual patient. The lower threshold of QR is limited by the PCR cycle at which mutation-free control samples show positivity. This depends on the exact mutation type and surrounding sequences that determine the primers and probes used for PCR. For example, in the case of *IDH2*R140Q mutation, the control samples become positive at the 32nd cycle (corresponding to 65 copies), while in the case of NPM1 mutation, only at the 42nd cycle (corresponding to 0.5 copies). This difference translates roughly to 3 logs. The upper threshold of QR is limited by the disease and mutation burden at diagnosis. Consequently, in a patient with *IDH2*R140Q mutation and 10,000 copies in the diagnostic sample, the QR would be approximately 2.5 logs. On the other hand, in a patient with NPM1 mutation and 2000 copies in a diagnostic sample, the QR would be more than 4 logs. Finally, differences between labs and PCR techniques does not allow wide standardization and incorporation of these methods into large clinical trials.

To overcome these hurdles, we harnessed the widely accepted commercial *ABL* standards by generating a plasmid that harbors both the sequence of interest and the *ABL* gene. The synthetic individual plasmid standards allow standardized results across labs and centers. In addition, copy numbers of those sequences per plasmid molecule are equivalents. Thus, the calculation of individual mutations’ copy number according to reference gene copies detected with commercial standard curve is possible. Results are reported both in change in copy number at various time points along the treatment course and in the depth of response and could be easily compared between labs. We provide our experience with this this approach in 19 AML patients with mutations in atypical *NPM1*, *IDH1/2* and *RUNX1* mutation. In all cases, there was a clear correlation between response and MRD levels. Most importantly, MRD monitoring of atypical *NPM1* and *RUNX1* mutations detected relapse, while still in CR by flow cytometry, and guided the clinical management.

## 5. Conclusions

Taken together, current standardized MRD assessment is limited to typical *NPM1* AML and CBF translocations, leaving most AML patients without a reliable molecular MRD marker. Our approach opts for in-house qPCR for individual mutation with plasmid-based synthetic standards by conjugating the mutation sequence to ABL sequence, allowing to report the result in copy number with validated and accepted ABL standards. This in turn opens the way to multi-center trials assessing the clinical significance of MRD monitoring of various mutations and MRD-based adaptive therapy.

## Figures and Tables

**Figure 1 cancers-15-05360-f001:**
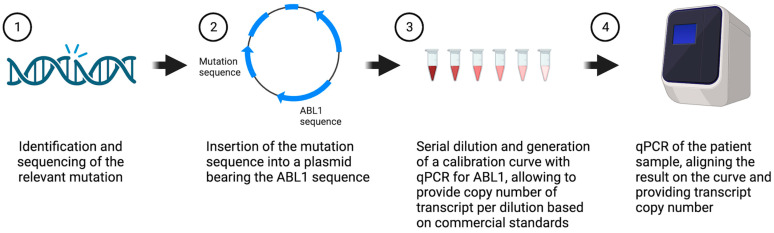
Synthetic gene method scheme. The identified sequence of the mutation of interest was inserted with the *ABL1* sequence into a plasmid in a 1:1 ratio. This allows us to translate the quantitative PCR (qPCR) result of the patient sample to a transcript copy number, using the *ABL1* commercial standards. Created with Biorender.com. https://www.biorender.com/, accessed on 7 November 2023.

**Figure 2 cancers-15-05360-f002:**
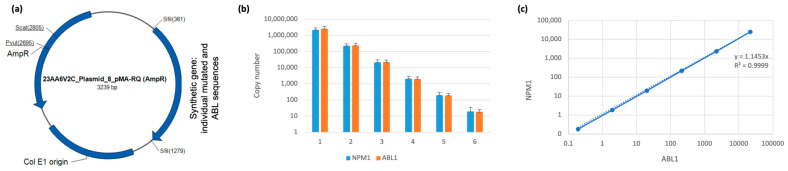
Generation and validation of synthetic gene. (**a**) *NPM1* type A and *ABL1* reference sequences were synthesized and cloned as a synthetic gene into standard vector. (**b**) Copy numbers for serial dilution of plasmid with synthetic sequences were defined by commercial standards for *NPM1* and *ABL1* in two separate reactions simultaneously in the same qPCR run. (**c**) Linear regression between *NPM1* and *ABL1* copy numbers was plotted.

**Figure 3 cancers-15-05360-f003:**
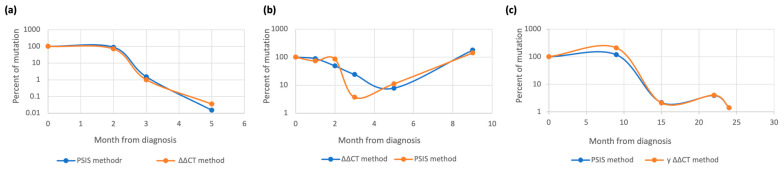
Determination of MRD status by ∆∆CT and copy number quantification: (**a**) *RUNX1*:c.663delinsGGA p.Phe221LeufsTer17 mutation; (**b**) *IDH2* R140Q mutation; (**c**) *RUNX1*: c.820del p.Gln274AsnfsTer37 mutation MRD status was defined by ∆∆CT with efficiency correction and specific individual synthetic sequence methods. The ∆∆CT with efficiency correction method allowed calculation of mutation percent for current sample versus diagnostic sample, as did the pSIS method based at quantitation by mutation-specific individual synthetic sequence. To compare results obtained by those methods, we converted data obtained with pSIS method in percent from diagnosis.

**Figure 4 cancers-15-05360-f004:**
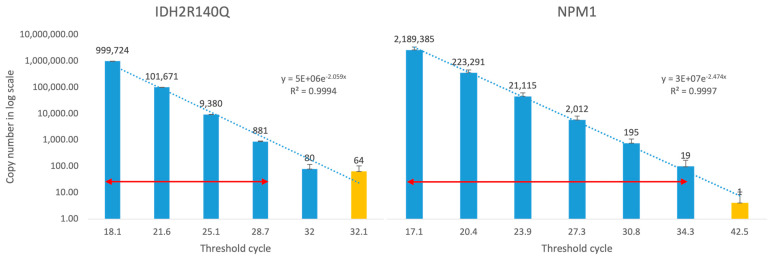
Quantitation range calculation. Quantitation range of individual synthetic standard is dependent on the mutation type, PCR primers and probe used in the method, as well as the disease burden at diagnosis. Blue bars represent mutated sequence and orange bars represent negative controls. Red arrows mark the quantitative range.

**Figure 5 cancers-15-05360-f005:**
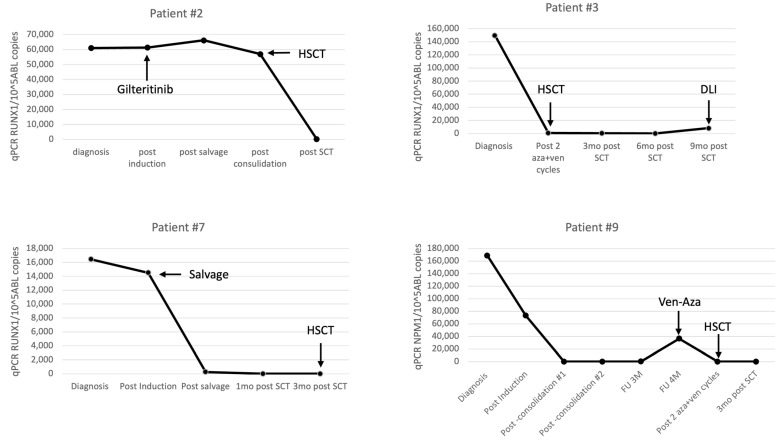
Copy number variation along the clinical course of selected patients. HSCT—hematopoietic stem cell transplant. FU—follow-up. DLI—donor lymphocyte infusion.

**Table 1 cancers-15-05360-t001:** Patient characteristics.

#	Sex	ECOG	Age	BM Blasts %	WBC Dx	Karyo-Type	Co-Mutations	DN/Sec	Plasmid Mutation	Induction	Salvage/ Consolidation	BMT	Death Cause	FU
1	M	1	61	1.7	1.5	−7Q	TET2 DNMT3 FLT3	Sec-TR	*Runx1*	azacitidine	pevonidestat	no	MM	30 m
2	M	2	72	22	4.30	Nk	TET2 SRSF2 SF3B1	DN	*Runx1*	Aza + ven	gilteritinib	MUD		12 m
3	M	1	66	8	1.50	Nk	BCORL1 SRSF2	Sec-MDS	*Runx1*	Aza + ven		MRF		3 m
4	F	0	59	40	3.5	t3:3	ASXL1 SF3B1	DN	*Runx1*	7 + 3	HAM	MUD		36 m
5	F	1	91	22	65	Nk	ASXL1 SRSF2 FLT3	DN	*Runx1*	Aza + ven		no		13 m
6	M	1	74	70	2.5	Tetra- ploid	FLT3 IDH1 PHF6 SRSF2	DN	*Runx1*	Aza + ven		no		10 m
7	F	0	39	20	7.1	Nk	BCOR WT1 DNMT3A KIT	DN	*Runx1*	7 + 3 + mido	flag + ida + ven	MRM		9 m
8	M	2	76	50	0.9	iso14	BCOR SRSF2 TET2	DN	*Runx1*	Decitabine (trial)r	Aza + ven	no		82 m
9	F	0	55	65	77	Nk	CEBPA TET2 MYC	DN	*NPM1*	7 + 3	HIDAC	HAP	acute GVHD	19 m
10	F	0	48	30	9.9	t3:11	FLT3	DN	*NPM1*	7 + 3 + mido	HIDAC	MUD		16 m
11	F	2	42	30	1.90	Nk	PTPN11 TET2	Sec-TR	*NPM1*	Aza + ven		MUD		24 m
12	F	0	19	80	1.30	Nk	None	DN	*NPM1*	7 + 3	HIDAC	MUD		58 m
13	M	0	24	60	1.5	Nk	None	DN	*NPM1*	7 + 3 + mido	HIDAC	no		35 m
14	M	2	81	45	1.9	Nk	ASXL1 STAT3 STAG2 SRSF2	Sec-MDS	*IDH2*	Aza + ven		no	Re-AML	15 m
15	M	0	60	45	9	tri11	U2AF1 DMNT3A	DN	*IDH2*	7 + 3	flag + ida + ven	MUD		10 m
16	F	2	62	70	3.7	mon7	BCOR DNMT3A EZH2	DN	*IDH2*	Aza + ven		MRM		17 m
17	M	2	69	50	1.4	Nk	RUNX1	DN	*IDH2*	Aza + ven		no		22 m
18	F	1	57	10	1.6	Nk	JAK2 MPL DMNT3A	Sec-ET	*IDH2*	Aza + ven		MUD		15 m
19	M	1	77	85	2.4	tri8	SRSF2 BCOR RUNX1	DN	*IDH1*	Aza + ven +/− magro (trial)		no		4 m

Sec TR—secondary treatment related, Sec MDS—Secondary to myelodysplastic syndrome.

**Table 2 cancers-15-05360-t002:** MRD monitoring results in patients referred to allo-BMT.

*RUNX1* Patient		Diagnosis	Post Induction	Post Salvage		Post Consolidation		Post-Transplant 1 m	FU 3 M	FU 6 M	FU 12 M
2	IP	40%	10%	4%		3.1%			1%	1.5%	
	STR								BM 100%	BM 100%	
	RUNX1 qPCR CN/10^5^ABL	61,023	61,369	66,152		56,979				315	
4	IP	40%	40%	2.6%		2.2%		2.6%	1.6%	1.4%	1.5%
	STR							BM 100%	BM 100%	BM 97%	BM 100%
	RUNX1 qPCR CN/10^5^ABL	436,945						neg	neg	neg	neg
**7**	IP	20%	15%	1.4%				1.4%	1.4%		
	STR							BM 100%	BM 100%		
	*RUNX1* qPCR CN/10^5^ABL	16,459	14,505	260				neg	neg		
***NPM1* Patient**		**Diagnosis**	**Post Induction**	**Post Consol.**	**Post Consol.**	**FU 3 M**	**FU 4 M**	**Post** **MRD Eraser cy 1**	**Post** **MRD Eraser cy 2**	**Post-Transplant 1 m**	**FU 3M**
9	IP	65%	1.7%	1.8%	2.4%	1.5%	2%	2%	0.8%		2.7%
	STR										BM 58%
	*NPM1* qPCR CN/10^5^ABL	168,717	73,274	68	45	258	36,362		39		neg
10	IP	30%	3%	0.6%				1.8%		1.2%	1%
	STR									BM 100%	BM 100%
	*NPM1* qPCR CN/10^5^ABL	1,038,889	15,557	239				79		neg	neg
***IDH* Patient**		**Diagnosis**	**Post Induction**	**Post Consol./Salvage**		**Post-Transplant 3 m**		**FU 6 M**		**FU 9 M**	**FU 12 M**
15	IP	45%	10%	1.5%		2%					
	STR					BM 100%					
	*IDH2* qPCR CN/10^5^ABL	96,964		967		neg					
16	IP	35%	3.3%	5%		1.8%					1.4%
	STR					BM 100%					BM100%
	*IDH2* qPCR CN/10^5^ABL	45,308	neg	106		neg					neg
18	IP	10%	2.5%	2.2%		0.8%		1.8%		1.2%	
	STR					BM 98%		BM 100%		BM 100%	
	*IDH2* qPCR CN/10^5^ABL	8911	50,659	neg		neg		70		neg	

IP—immunophenotyping; FU—follow-up; consol.—consolidation.

**Table 3 cancers-15-05360-t003:** MRD monitoring results in patients not referred to allo-BMT.

Patient		Diagnosis	Post Induction/ 1st Aza + Ven Cyc	Post Consol.	Post Consol.	FU 3 M/ Post 3 Aza + Ven Cycles	FU 6 M/ Post 6 Aza + Ven Cycles	FU 8 M/ Post 8 Aza + Ven Cycles	FU 12 M/ Post 12 Aza + Ven Cycles	FU 24 M/ Post 24 Aza + Ven Cycles
13	IP	60%	4%	1.9%	2.3%	2.1%	2%		1.3%	1.5%
	*NPM1* qPCR CN/10^5^ABL	1,414,578	216	45	40				3.2	0.76
17	IP	50%				2.5%	2.6%	1.1%	1.5%	0.5%
	*IDH2* qPCR CN/10^5^ABL	115,521				57,677	4681	2149	2243	403

IP—immunophenotyping; FU—follow-up; consol.—consolidation.

**Table 4 cancers-15-05360-t004:** MRD monitoring results in a relapsed patient.

Patient		Diagnosis	Post Induction	Post Consol.	FU 12 M	FU 24 M	FU 30 M	FU 33 M	FU 36 M	FU 37 M	Post Salvage	FU 6 M
12	IP	63%	3.5%	1.6%	2.5%	2.6%	3%	10%	18%	26%	1%	2%
	STR											BM 100%
	*NPM1* qPCR CN/10^5^ABL	3,221,400	49	neg	neg	314	98,700	463,417	529,710	828,789	neg	neg

IP—immunophenotyping; FU—follow up; consol.—consolidation.

## Data Availability

All data are available in the manuscript.

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
