# Peer review of "Standardization of Molecular MRD Levels in AML Using an Integral Vector Bearing ABL and the Mutation of Interest"

_cancers, 2023, doi:10.3390/cancers15225360_

Round 1

Reviewer 1 Report

Comments and Suggestions for Authors

This is a nice study designed by the authors for in-house MRD monitoring for identified mutations using minigene constructs. This is a great approach and I am impressed with the work. I only have minor concerns regarding the overall presentation of the research article:

1. In the Introduction part a few lines about the importance of MRD and mutation-based MRD detection techniques should be added for reader clarity.

2. Limitations and advantages of standardized qPCR techniques for MRD should be explained.

3. The key message of the work being done should reiterated.

4. Overall methods part is a bit complicated to understand in the first go, I think it can be put in more detail which will make it easy to understand. e.g. Describe the PCR system more clearly, and practical implementation of individual systems for patients.  

5. The addition of a single figure or flowchart describing the complete methods part would also be helpful in visually understanding the process.

6.  There are places where acronyms are not defined at their first usage. Please define them.

7. You might consider using subheadings within the results section to separate different aspects of your findings, such as "Outcomes in Patients with RUNX1 Mutations" or "IDH Monitoring Outcomes."

8. Overall, your discussion is informative, but it could benefit from providing more detailed insights into the clinical significance, limitations, and future prospects of your approach to MRD monitoring. Additionally, providing a clear and concise summary of your main findings can help readers grasp the significance of your work.

Comments on the Quality of English Language

The use of the English language is fine. I do not see any major errors in the language usage. 

Reviewer 2 Report

Comments and Suggestions for Authors

This study by Nachmias et al is an interesting summary of a plasmid-based MRD assay for common AML mutations affecting NPM1, IDH1/2 and RUNX1. Some comments should be addressed:

1. introduction: please explain why the authors focused on NPM1, IDH1/2 and RUNX1. Several other genes commonly mutated in AML could have been chosen. The rational should be better explained. 

2. Results: few patients (#6, 17 and 19)showed concomitant mutations in more than 1 of the assay genes. The authors should test for all mutations i these patients to directly compare the sensitivity of the genes in a real-life setting.

3. Discussion: the issue of mutation persistence after CR without a clear clinical consequence should be discussed in more detail (see PMID: 29601269PMID: 30610028, PMID: 29472724, and others).

4. Discussion: the approach is somehow outdated by error-corrected NGS-approaches. Please discuss this in more detail

5. minor comment: all gene names should be written in italics in text, tables, and figures   

Round 2

Reviewer 2 Report

Comments and Suggestions for Authors

No further comments